# Keeping Pathologists in the Loop and an Adaptive F1-Score Threshold Method for Mitosis Detection in Canine Perivascular Wall Tumours

**DOI:** 10.3390/cancers16030644

**Published:** 2024-02-02

**Authors:** Taranpreet Rai, Ambra Morisi, Barbara Bacci, Nicholas James Bacon, Michael J. Dark, Tawfik Aboellail, Spencer A. Thomas, Roberto M. La Ragione, Kevin Wells

**Affiliations:** 1Centre for Vision, Speech and Signal Processing, University of Surrey, Guildford GU2 7XH, UK; k.wells@surrey.ac.uk; 2Surrey DataHub, University of Surrey, Guildford GU2 7AL, UK; 3School of Veterinary Medicine, University of Surrey, Guildford GU2 7AL, UK; ambramorisi@hotmail.it (A.M.); r.laragione@surrey.ac.uk (R.M.L.R.); 4Department of Veterinary Medical Sciences, University of Bologna, 40126 Bologna, Italy; barbara.bacci@gmail.com; 5AURA Veterinary, Guildford GU2 7AJ, UK; n.bacon@surrey.ac.uk; 6Department of Comparative, Diagnostic and Population Medicine, College of Veterinary Medicine, University of Florida, Gainesville, FL 32611, USA; darkmich@ufl.edu; 7Department of Diagnostic Pathology and Pathobiology, Kansas State University, Manhattan, KS 66506, USA; tawfik.aboellail@colostate.edu; 8Department of Computer Science, University of Surrey, Guildford GU2 7XH, UK; spencer.thomas@npl.co.uk; 9National Physical Laboratory, London TW11 0LW, UK; 10School of Biosciences, University of Surrey, Guildford GU2 7XH, UK

**Keywords:** artificial intelligence, deep learning, canine Soft Tissue Sarcoma, canine Perivascular Wall Tumour, digital pathology, object detection, faster R-CNN, mitosis, mitosis detection, pathologists in the loop, humans in the loop

## Abstract

**Simple Summary:**

Performing a mitosis count (MC) is essential in grading canine Soft Tissue Sarcoma (cSTS) and canine Perivascular Wall Tumours (cPWTs), although it is subject to inter- and intra-observer variability. To enhance standardisation, an artificial intelligence mitosis detection approach was investigated. A two-step annotation process was utilised with a pre-trained Faster R-CNN model, refined through veterinary pathologists’ reviews of false positives, and subsequently optimised using an F1-score thresholding method to maximise accuracy measures. The study achieved a best F1-score of 0.75, demonstrating competitiveness in the field of canine mitosis detection.

**Abstract:**

Performing a mitosis count (MC) is the diagnostic task of histologically grading canine Soft Tissue Sarcoma (cSTS). However, mitosis count is subject to inter- and intra-observer variability. Deep learning models can offer a standardisation in the process of MC used to histologically grade canine Soft Tissue Sarcomas. Subsequently, the focus of this study was mitosis detection in canine Perivascular Wall Tumours (cPWTs). Generating mitosis annotations is a long and arduous process open to inter-observer variability. Therefore, by keeping pathologists in the loop, a two-step annotation process was performed where a pre-trained Faster R-CNN model was trained on initial annotations provided by veterinary pathologists. The pathologists reviewed the output false positive mitosis candidates and determined whether these were overlooked candidates, thus updating the dataset. Faster R-CNN was then trained on this updated dataset. An optimal decision threshold was applied to maximise the F1-score predetermined using the validation set and produced our best F1-score of 0.75, which is competitive with the state of the art in the canine mitosis domain.

## 1. Introduction

Canine Soft Tissue Sarcoma (cSTS) is a heterogeneous group of mesenchymal neoplasms (tumours) that arise in connective tissue [1,2,3,4,5,6]. cSTS is more prevalent in middle-age to older and medium to large-sized breeds with the median reported age of diagnosis between 10 and 11 years old [3,7,8,9,10]. The anatomical site of cSTS can vary considerably, but it is mostly found in the cutaneous and subcutaneous tissues [9]. In human Soft Tissue Sarcoma (STS), histological grade is an important prognostic factor and one of the most validated criteria to predict outcome following surgery in canines [10,11,12,13]. General treatment consists of surgically removing these cutaneous and subcutaneous sarcomas. Nevertheless, it is the higher-grade tumours that can be problematic, as their aggressiveness can reduce treatment options and result in a poorer prognosis. The focus of this study was on one common subtype found in dogs: canine Perivascular Wall Tumours (cPWTs). Canine Perivascular Wall Tumours (cPWTs) arise from vascular mural cells and are often recognisable from their vascular growth patterns [14,15].

The scoring for cSTS grading is broken down into three major criteria: the mitotic count, differentiation and the level of necrosis [9]. Mitosis counting can be exposed to high inter-observer variability [16], depending on the expertise of the pathologist; however, the counting of mitotic figures is considered the most objective factor in comparison to tumour necrosis and cellular differentiation when grading cSTS [16]. It is routine practise to investigate mitosis using 40× magnification; however, manual investigation at such high-powered fields (HPFs) is a laborious task that is prone to error, thus leading to the previously discussed inter-observer variability phenomenon.

For the purposes of this study, the focus was on creating a mitosis detection model as it is a significant criterion from the cSTS histological grading system [13] where the density of mitotic figures is also considered highly correlated with tumour proliferation [17]. Mitosis detection has been pursued in the computer vision domain since the 1980s [18]. Before 2010, relatively few studies aimed to automate mitosis detection [19,20,21]. However, since the MITOS 2012 challenge [22], there has been a resurgence of interest. Mitosis detection can often be considered as an object detection problem [23]. Rather than categorising entire images as in image classification tasks, object detection algorithms present object categories inside the image along with an axis-aligned bounding box, which in turn indicates the position and scale of each instance of the object category. In the case of mitosis detection, the considered objects are mitotic figures. As a result, several approaches have used object detection-related algorithms for mitosis detection. An example of an object detection algorithm is the regions-based convolutional neural network (R-CNN) [24]. At first, a selective search is performed on the input image to propose candidate regions, and then the CNN is used for feature extraction. These feature vectors are used for training in bounding box regression. There have been many developments on this type of architecture such as Fast R-CNN [25] and Faster R-CNN [26], which is the primary object detection model used in this work. One set of authors detected mitosis using a variant of the Faster R-CNN (MITOS-RCNN), achieving an F-measure score of 0.955 [27].

Several challenges have been held in order to find novel and improved approaches for mitosis detection [17,22,23,28,29]. Some of these challenges and research on mitosis detection methods have also been conducted using tissue from the canine domain [30,31,32,33].

It was made apparent by the collaborating pathologists that AI approaches for grading tasks in cSTS were desirable, and so this study aims to tackle one criterion, which is to develop methods for mitosis detection in a subtype of cSTS: cPWT. To the best of our knowledge, this is the first work in the automated detection of mitoses in cPWTs.

## 2. Materials and Methods

### 2.1. Data Description and Annotation Process

A set of canine Perivascular Wall Tumour (cPWT) slides were obtained from the Department of Microbiology, Immunology and Pathology, Colorado State University. A senior veterinary pathologist at the University of Surrey confirmed the grade of each case (patient) and chose a representative histological slide for each patient. These histological slides were digitised using a Hamamatsu NDP Nanozoomer 2.0 HT slide scanner. A digital Whole Slide Image (WSI) was created via scanning under 40× magnification (0.23 µm/pixel) with a scanning speed of approximately 150 s at 40× mode (15 mm × 15 mm).

Veterinary pathologists independently annotated the WSIs for mitosis using the open-source Automated Slide Analysis Platform (ASAP) software (https://www.computationalpathologygroup.eu/software/asap/, accessed on 28 January 2024) [34]. The pathologists used different magnifications (ranging from 10× to 40×) to analyse the mitosis before creating mitosis annotations. These annotations were centroid coordinates, which were centered on the suspecting mitotic candidate. Centroid coordinate annotations can be considered as weak annotations as they are simply coordinates placed in the centre of a mitotic figure and not fine-grained pixel-wise annotations around the mitosis. In order to categorise a mitotic figure, both pathologist annotators needed to form an agreement on the mitotic candidate. As these were centroid coordinates, an agreement was determined when two independent centroid annotations from each annotator were overlaid on one another. Any centroid annotations without agreement were dismissed from being considered as a mitotic figure. Table 1 shows the differences between the two annotators for both training and validation when counting mitotic figures in our cPWT dataset.

For patch extraction, downsized binary image masks (by a factor of 32) were generated, depicting tissue from the biopsy samples against background slide glass. A tissue threshold of 0.75 was applied to 512 × 512 patches for final patch extraction. Therefore, if a patch contained less than 75% of any tissue, it was dismissed from the dataset. This was to ensure that the patches contained relevant information for mitosis object detection.

The test set consisted of patches extracted from high-powered fields (HPFs) determined by the pathologist annotators. To replicate real-world test data, our collaborating pathologists selected 10 continuous non-overlapping HPFs from each WSI. The size of this area was determined by loosely following the Elston and Ellis [35] criteria of an area size of 2.0 mm^2^. For 20× magnification (level 1 in the WSI pyramid), the width of the 10 HPFs was 4096 pixels and the height was 2560 pixels. This produced 40 non-overlapping patches of 512 × 512 pixels, thus producing a dataset of 440 patch images from the 11 hold-out test WSIs at 20× magnification. Only patches containing mitosis were used for training and validation, whereas for testing, all extracted patches were evaluated. Details on the number of mitosis per slide in training/validation and test sets are provided in Appendix A
Table A1 and Table A2, respectively. Details on the number of patches used for training/validation and testing for 40× magnification is provided in Appendix A
Table A3. Details on the number of patches used for training/validation and testing for 20× magnification is provided in Appendix A
Table A4.

### 2.2. Object Detection and Keeping the Pathologist in the Loop for Dataset Refinement

Mitosis detection is generally considered an object detection problem [23]; For this study, we used a Faster R-CNN model [26]. We initialised a Faster R-CNN model with pre-trained COCO [36] weights with the ResNet-50 head pre-trained on ImageNet. The model was fine-tuned, updating all parameters of the model using our dataset. Preliminary experiments suggested using a learning rate of 0.01 and SGD to be used as the optimiser. A batch size of 4 was also used for these experiments. Training was implemented for 30 epochs, where the the model with the lowest validation loss was saved for final evaluation. Faster R-CNN is jointly trained with four different losses; two for the RPN and two for the Fast R-CNN. These losses are RPN classification loss (for distinguishing between foreground and background), RPN regression loss (for determining differences between the regression of the foreground bounding box and ground truth bounding box), the Fast R-CNN classification loss (for object classes) and Fast R-CNN bounding box regression (used to refine the bounding box coordinates). Therefore, in our implementation of determining the lowest validation loss, at every epoch, each loss type was considered equally. We implemented 3-fold cross-validation at the patient (WSI) level to test the veracity and robustness of our approach with the training data split into three folds for training and validation. We also used an unseen hold-out test set for final evaluation and for a fair comparison of all three folds. The training, validation and hold-out test splits for each fold are depicted in Appendix A
Table A5.

Furthermore, as most mitotic figures from the same tissue type are generally of a similar size (dependent on the stage of mitosis, staining techniques, and slide quality), we opted to use the default anchor generator sizes provided by the PyTorch implementation of Faster R-CNN. These sizes were 32, 64, 128, 256 and 512 with aspect ratios of 0.5, 1.0 and 2.0. See Figure 1 for a depiction of the Faster R-CNN applied to the cPWT mitosis detection problem.

During the evaluation inference, non-maximum suppression (NMS) with an IoU value of 0.1 was applied as a post-processing step to remove low-scoring otherwise redundant overlapping bounding boxes. This post-processing method is also consistent with other mitosis detection methods in the literature [38,39].

In object detection, mean average precision (mAP) is typically used to evaluate the performance of a model depending on the task or dataset [40,41,42,43]. However, we opted to use the F1-score in order to compare our results to mitosis detection approaches in the literature. The F1-score was computed globally for each fold; thus, it was applied and determined for the entire dataset of interest. True positive (TP) detections were computed if there was an IoU of >= 0.5 between the ground truth and proposed candidate detections. Anything that did not meet the IoU threshold was considered a false positive (FP) detection. Any missed ground truth detections were considered false negatives (FNs). As a result, we could also generate the F1-score. The F1-score can be considered the harmonic mean between the precision and recall (sensitivity). Both precision (Equation (Equation 1)) and sensitivity (Equation (Equation 2)) contribute equally to the F1-score (Equation (Equation 3)):(1)$$\begin{array}{c} Precision=\frac{TP}{TP+FP} \end{array}$$
(2)$$\begin{array}{c} Sensitivity=\frac{TP}{TP+FN} \end{array}$$
(3)$$\begin{array}{c} F1=2*\frac{Sensitivity*Precision}{Sensitivity+Precision} \end{array}$$
where TP, FP and FN are true positives, false positives and false negatives, respectively.

The models were implemented in Python, using the PyTorch deep learning framework. The hardware and resources available for implementation used a Dell T630 system, which included 2 Intel Xeon E5 v4 series 8-Core CPUs with 3.2 GHz, 128 GB of RAM (Dell Corporation Limited, London, UK), and 4 nVidia Titan X (Pascal, Compute 6.1, Single Precision) GPUs.

The mitosis annotation process is an exhaustive and arduous process, and thus the initial annotation process may be suboptimal due to the vast area annotators needed to examine mitotic candidates. Taking inspiration from Bertram et al. [33], we used our deep learning object detection models from these experiments to refine the dataset (see Figure 2). We hypothesised that many of the FP candidates may have been incorrectly labelled. Our collaborating pathologists reviewed all the FP candidates (irregardless of class score) from each validation fold and the hold-out test set and determined which candidates were mislabeled. As a result, we were able to formulate additional ground truth mitoses for use in the final set of experiments.

### 2.3. Adaptive F1-Score Threshold

For this method, the Faster R-CNN object detector was trained on detecting mitotic candidates using the refined (updated) dataset. The same training hyperparameters as described earlier were applied; however, we lowered the number of epochs. It was observed that the models found their optimal validation loss by epoch 7 across all three folds in the initial experiment runs. Therefore, to ensure optimality, we chose 12 epochs for training, again using the lowest validation loss as determining the “best” model. The trained Faster R-CNN model outputs potential mitosis candidates, but it also outputs probability scores relating to the strength of the object prediction. These scores ranged from 0 to 1, where 1 would highlight the model is 100% certain that the candidate is mitosis and 0.01 would describe a prediction that is very low in confidence. We optimised our models based on the F1-score [44,45,46]. The probability thresholds *t* ranged from 0.01 to 1, and so choosing the optimal threshold *T* for the F1-score F1 can be represented formally as:(4)$$\begin{array}{c} T=\underset{t}{arg max}F1\left(t\right) \end{array}$$

We determined the optimal F1-score threshold value using the validation set and applied this threshold value to the final evaluation on the hold-out test set. Figure 3 demonstrates the entire workflow of this method from the creation of the updated mitosis dataset where the pathologists reviewed all the FP candidates all the way to the adaptive F1-score thresholds applied to the mitosis candidate predictions.

## 3. Results

The pathologists-in-the-loop approach for dataset refinement was first applied as demonstrated by Figure 2. In a preliminary investigation, two magnifications (40× and 20×) were used to determine the best resolution for our for our task (see Table 2).

Table A6 and Table A7 show the differences in mitotic candidate numbers before and after refinement (second review) for the training/validation and test sets, respectively. The first set of results from the optimised Faster R-CNN approach is depicted in Table 3. This shows a comparison of performance of the Faster R-CNN trained on the initial mitosis dataset and the updated refined mitosis dataset. It is apparent that sensitivities have improved for all folds when using the updated refined dataset; however, in some cases, such as in fold-1 validation, fold-3 validation and fold-3 test, we can see that the F1-score is lower due to a decrease in precision scores. This could be due to the updated refined dataset containing more difficult examples for the effective mitosis object detection training. The previous initial dataset may have contained more obvious mitosis examples and thus was predicting detections that closely resembled these obvious examples. Table 4 shows the Faster R-CNN results before and after F1-score thresholding was applied on the models trained using the updated mitosis dataset. The thresholds were predetermined on the validation set for each fold using Equation (Equation 4) (see Figure 4). When applying the optimal thresholds, we saw large improvements in the F1-score, which were largely due to an improvement in precision because of a reduction in FPs. This was seen on the test set with an F1-score of 0.402 to 0.750. However, this increase in precision came at the expense of some sensitivity across all three folds, where for example on the test set the mean sensitivity for all three folds reduced from 0.952 to 0.803. Nevertheless, the depreciation in sensitivity does not offset the increase in precision, where sensitivity decreased by 14.9 % and precision increased by 45.2 %. This suggests that the majority of TP detections prior to the adaptive F1-score thresholding are of a high probability confidence compared to the FP detections.

## 4. Discussion

This study has demonstrated a method for mitosis detection in cPWT WSIs using a Faster R-CNN object detection model, an adaptive F1-score thresholding feature on output probabilities and the refinement of a mitotic figures dataset by keeping pathologists in the loop.

Many approaches in the literature use the highest resolution images for their object detection methods (typically at 40× objective); however, we preliminarily found that 20× magnification was beneficial for our task and the dataset provided, as shown in Table 2. Nevertheless, this warrants a further investigation and additional discussions with the collaborating pathologists, who may provide reasoning as to why certain candidates were classed as mitosis at different resolutions.

Initially, solely using the outputs from a Faster R-CNN model produced promising results generating high sensitivities; however, these outputs required further post-processing to improve precision. Applying adaptive F1-score thresholds, where the optimal values were predetermined on the validation set and applied to the test set, demonstrated an effective method of reducing the number of FP predictions. This ultimately resulted in dramatically increasing the F1-score due to a stark increase in precision. However, this came at a small expense of sensitivity. Nevertheless, the rate of change of the sensitivity and the precision are not equal with the latter vastly improving. This suggests that the majority of FP detections are of lower probability confidence compared to TP detections.

Multi-stage (typically dual-stage) approaches have also become increasingly prevalent over the years where they typically take the form of selecting mitotic candidates in the first stage and then apply another classifier in the second stage [32,33,47,48,49]. Although not reflected in the main findings of this study, we attempted to use a second-stage classifier (Figure A1) on mitotic candidates to classify between TP and hard FPs to no avail (see results of the two-stage approach in Table A8 and its subsequent ROC curves in Figure A2). Most machine learning methods require large datasets for effective training, which in this case was not available once optimisation was applied using the adaptive F1-score threshold method. One could train models using the non-thresholded detections; however, this would result in a model that is able to distinguish between true positive mitosis and mostly obvious FP candidates. By applying the adaptive F1-score thresholding method, we constrained the dataset and attempted to learn differences between TP and high confidence hard false positive detections, but we did not provide an adequately large dataset for training. Figure 5 depicts a 512 × 512 pixel image in the test set, highlighting FN and FP detection.

Different phases and other biological phenomenon could influence the size of the mitosis region of interest. Going forward, it may also be worth labelling mitosis in regard to the phases and thus creating a multi-class problem rather than binary, as shown in this study. As a consequence, the size of the ground truth bounding boxes could also be varied depending on the target phase being classified. Nonetheless, the models were still able to predict the vast majority of mitosis in these phases.

It must be further denoted that the methodology is applied to only patches from HPFs containing mitosis that were annotated by the collaborating pathologists. Therefore, we propose expanding our dataset to include a broader range of sections, including those not initially marked by pathologists, to evaluate and enhance our model’s generalisability. The data should include labels for areas containing tumour and non-tumour tissue to fully consider the overall impact of this mitosis detection method.

Our focus for this study is on cPWT; however, we could potentially adapt this method to other cSTS subtypes as well as to other tumour types. An additional study might explore the application of cPWT-trained models to different cSTS subtypes to assess if comparable outcomes are achieved. Nevertheless, given that tumour types from various domains exhibit unique challenges due to their specific histological characteristics, it may be necessary to train or fine-tune models using tumour-specific datasets to evaluate the efficacy of this approach.

While our F1-score demonstrates competitive performance for detecting mitosis in the canine domain, the clinical relevance and applicability of this metric should be taken into account. Future work should focus on employing this method as a supportive tool, assessing its practical effectiveness and reliability in a veterinary clinical setting.

To conclude, by using our experimental set-up, the optimised Faster R-CNN model was a suitable method for determining mitosis in cPWT WSIs. To the best of our knowledge, this is the first mitosis detection model applied solely on cPWT data, and thus we consider this a baseline three-fold cross-validation mean F1-score of 0.750 for mitosis detection in cPWT.

## Figures and Tables

**Figure 1 cancers-16-00644-f001:**
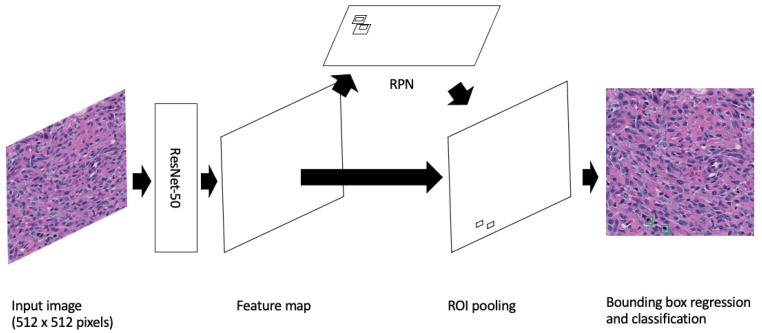
Image is inspired by Mahmood et al.’s depiction of Faster R-CNN [37]. A Faster R-CNN object detection model applied to the cPWT mitosis dataset. An input image of size 512 × 512 pixels is passed through the model where the feature map is extracted using the Resnet-50 feature-extraction network. This is then followed by generating region proposals in the Region Proposal Network (RPN) and finally mitosis detection in the classifier.

**Figure 2 cancers-16-00644-f002:**
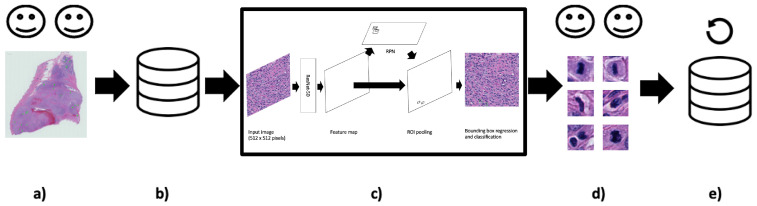
Keeping humans in the loop: (**a**) Two pathologist annotators independently review canine Perivascular Wall Tumour (cPWT) Whole Slide Images (WSIs) and applied centroid annotations to mitotic figures. (**b**) After initial agreement of mitoses, this formed the initial dataset of patch images with annotations. (**c**) A Faster R-CNN object detector was trained and tested on these data. (**d**) Thereafter, false positive (FP) candidates are reviewed again by the pathologist annotators where misclassified candidates are reassigned as true positives (TPs). (**e**) These new TPs are added to the updated dataset. (20× magnification images).

**Figure 3 cancers-16-00644-f003:**
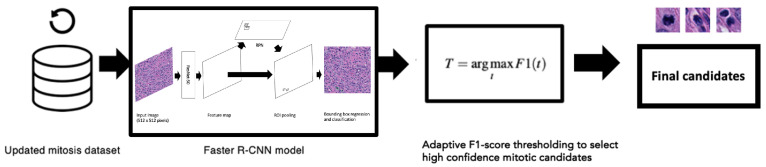
We used 20× magnification images and annotations from the updated mitosis dataset to train the Faster R-CNN object detection model (details from the Faster R-CNN model are also shown in Figure 1). Optimal thresholds using Equation (Equation 4) were applied on the output candidates determined from the validation set.

**Figure 4 cancers-16-00644-f004:**
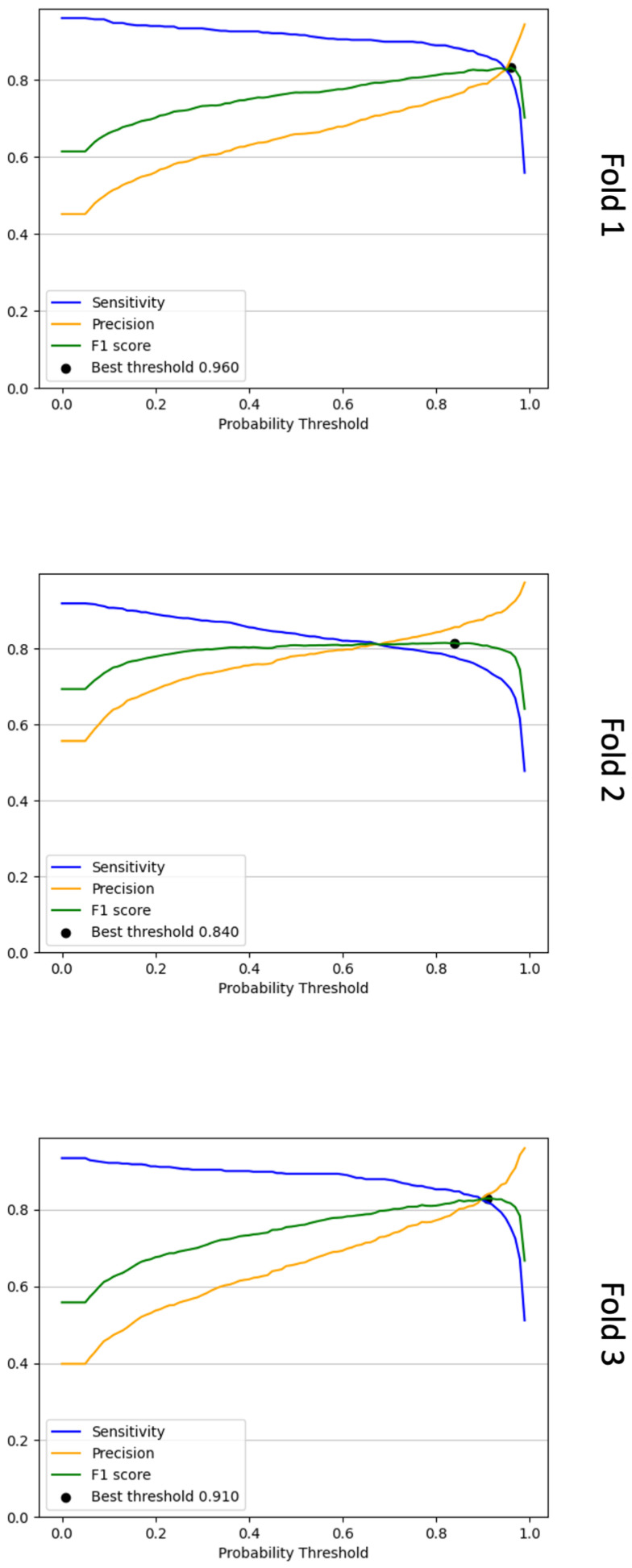
Line graphs that show the sensitivity, precision and F1-score calculated for each probability threshold for the three validation folds. To determine the optimal probability threshold, we choose the threshold with the highest F1-score as determined via Equation (Equation 4). In the above plots, these are denoted as “best threshold”. For fold 1, this threshold was 0.96, for fold 2, it was 0.84, and for fold 3, it was 0.91.

**Figure 5 cancers-16-00644-f005:**
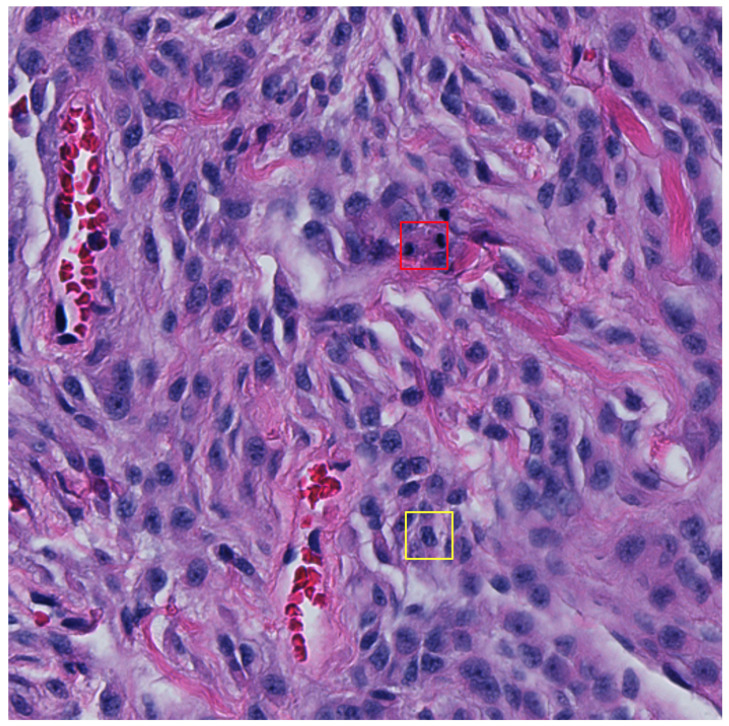
An example 512 × 512 pixel image from the test set with a false negative (FN) shown in the red bounding box and a false positive (FP) detection shown in the yellow bounding box (32 × 32 pixels). The FP detection provides a probability confidence score of 5.3% and so would typically be dismissed as a mitosis candidate once the adaptive F1-score threshold is applied.

**Table 1 cancers-16-00644-t001:** The differences between the two annotators in regard to mitosis annotations for the training/validation set. The “Slide” column represents the anonymised set of slides annotated. “Anno 1” and “Anno 2” show the number of mitoses annotated per slide for each annotator. “Agreement” represents the number of agreed mitoses between each annotator. The “% agreement” for each annotator represents the percentage of the agreed mitotic count against the respective annotators mitotic count. “Avg” is the average of every WSI % agreement, which is computed for each annotator.

Slide	Anno 1	Anno 2	Agreement	% Agreement Anno 1	% Agreement Anno 2
F17-04773	31	31	23	74.19	74.19
F17-03141	69	89	55	79.71	61.80
F17-1261	45	46	41	91.11	89.13
F18-13364	695	517	444	63.88	85.88
F17-02232	331	264	218	65.86	82.58
F17-04911	49	58	37	75.51	63.79
F17-0549	157	142	112	71.34	78.87
F17-011577	27	29	23	85.19	79.31
F17-011777	449	367	290	64.59	79.02
F17-03855	97	87	70	72.16	80.46
F17-04900	91	86	75	82.42	87.21
F18-7832	496	401	346	69.76	86.28
F17-09700	202	187	139	68.81	74.33
F17-02641	59	48	43	72.88	89.58
F17-09926	77	71	62	80.52	87.32
F17-02723	49	52	40	81.63	76.92
F17-05935	55	46	44	80.00	95.65
F17-02120	58	53	43	74.14	81.13
F18-79705	132	99	87	65.91	87.88
Total:	3169	2673	2192	Avg: 74.72	Avg: 81.12

**Table 2 cancers-16-00644-t002:** Initial mitosis object detection results for the 40× and 20× magnification patches datasets. As the difference in performance between the two resolution datasets was of interest, we first present the initial results for 20× and 40 magnifications for validation and test sets and for all three folds. Interestingly, although the 40× magnification trained models appeared to produce better F1-scores for validation, 20× magnification models performed better across all three folds when applied to the hold-out test set. It appears that with our experimental set-up, the models trained on 20× magnification generalise across the two evaluation datasets better. As a consequence, and to also reduce computational requirements, we proceeded further with the 20× magnification extracted dataset. Results for these initial experiments also suggested that the object detector was highly sensitive for the test set (at a mean average of 0.918) but not as precise (at a mean average of 0.249 for the precision measure).

Magnification	Fold	Set	Sensitivity	Precision	F1-Score	TP	FP	FN
40×	1	Val	0.967	0.720	0.826	590	229	20
40×	1	Test	0.957	0.132	0.232	135	890	6
40×	2	Val	0.922	0.786	0.849	847	230	72
40×	2	Test	0.965	0.173	0.294	136	649	5
40×	3	Val	0.944	0.724	0.819	503	192	30
40×	3	Test	0.957	0.185	0.311	135	593	6
20×	1	Val	0.957	0.484	0.643	582	620	26
20×	1	Test	0.932	0.207	0.338	137	526	10
20×	2	Val	0.895	0.567	0.694	810	619	95
20×	2	Test	0.918	0.221	0.356	135	477	12
20×	3	Val	0.897	0.545	0.678	477	399	55
20×	3	Test	0.905	0.320	0.473	133	282	14

**Table 3 cancers-16-00644-t003:** A comparison of results of the models trained on the initial annotated dataset and the updated dataset. Results are shown for both the validation and test sets for folds 1, 2 and 3.

Fold	Data	Set	Sensitivity	Precision	F1-Score	TP	FP	FN
1	Initial	Val	0.957	0.484	0.643	582	620	26
1	Updated	Val	0.961	0.452	0.615	610	740	25
1	Initial	Test	0.932	0.207	0.338	137	526	10
1	Updated	Test	0.954	0.239	0.383	187	594	9
2	Initial	Val	0.895	0.567	0.694	810	619	95
2	Updated	Val	0.919	0.557	0.694	877	698	77
2	Initial	Test	0.918	0.221	0.356	135	477	12
2	Updated	Test	0.959	0.281	0.435	188	480	8
3	Initial	Val	0.897	0.545	0.678	477	399	55
3	Updated	Val	0.935	0.398	0.558	528	798	37
3	Initial	Test	0.905	0.320	0.473	133	282	14
3	Updated	Test	0.944	0.244	0.387	185	574	11

**Table 4 cancers-16-00644-t004:** Results of the models trained on the updated dataset. The “Thresholds” column depict whether models were optimised using the adaptive F1-score threshold metric described in Equation (Equation 4); filled in values state the probability threshold. It is apparent that the models with optimised thresholds produced the highest F1-scores across all folds, producing a mean average F1-score of 0.750 on the test set compared to 0.402.

Fold	Threshold	Set	Sensitivity	Precision	F1-Score
1	None	Val	0.961	0.452	0.615
1	0.96	Val	0.811	0.854	0.832
1	None	Test	0.954	0.239	0.383
1	0.96	Test	0.776	0.756	0.766
2	None	Val	0.919	0.557	0.694
2	0.84	Val	0.778	0.857	0.815
2	None	Test	0.959	0.281	0.435
2	0.84	Test	0.827	0.633	0.717
3	None	Val	0.935	0.398	0.558
3	0.91	Val	0.819	0.840	0.830
3	None	Test	0.944	0.244	0.387
3	0.91	Test	0.806	0.731	0.767
Average (mean)	None	Val	0.938	0.469	0.622
		Test	0.952	0.255	0.402
	Optimised	Val	0.803	0.850	0.826
		Test	0.803	0.707	0.750

## Data Availability

The Whole Slide Images used in this study are available from the corresponding author on reasonable request. Code and annotations are currently unavailable.

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
