# Peer review of "Keeping Pathologists in the Loop and an Adaptive F1-Score Threshold Method for Mitosis Detection in Canine Perivascular Wall Tumours"

_cancers, 2024, doi:10.3390/cancers16030644_

Round 1

Reviewer 1 Report

Comments and Suggestions for Authors

In this manuscript, the authors describes "Keeping Pathologists-in-the-Loop and an Adaptive F1-Score Threshold Method for Mitosis Detection in Canine Perivascular Wall Tumours".

While is manuscript is well written, it would be helpful if the authors address these minor concerns.

1) In line 189, it would be helpful if figure 4 is placed on the same page as its description to prevent unnecessary distractions for readers.

2) While a depiction of a faster R-CNN object detection model (Figure 1), independently reviewed cPWT whole slide images (Figure 2) and magnified images and annotations from updated mitosis data set (Figure 3) is clearly illustrated, no reference of these figures were made in the manuscript. The authors should consider revising this.

In short, this manuscript could potentially benefit its target audience if the above concerns are addressed.

Author Response

Thank you for your insightful comments, and recommendations to improve the manuscript. We have taken into account your suggestions as follows:

  • We have moved figure 4 (which also takes up the entire page) closer to line 189 so that it is easier to navigate to for the reader.

  • Thank you for highlighting this very important point. We have now added text that refers back to these figures on line 128, 228 (Figure 1), 229 (Figure2) and line 176 (Figure 3) at the end of the “Adaptive F1-score threshold” section.

We have also amended some grammar in the introduction text for better flow and added additional relevant references from the canine mitosis domain to improve the background of this work.

Text added to the Introduction: “Some of these challenges and research on mitosis detection methods have also been conducted using tissue from the canine domain [1] [3] [4] [10].”

Reviewer 2 Report

Comments and Suggestions for Authors

This study aimed to establish an AI model for automatically detecting mitotic figures in canine perivascular wall tumors. Several problems exist for the clinical application of this method, which have not been fully considered.

For example, whether mitotic figures can be detected only in representative sections, what should be done when tumor and non-tumor areas exist in one section, whether the F-score obtained is worthy of clinical application, whether similar results can be obtained in other tumors, and so on. At a minimum, additional discussion of such points is needed.

In addition, studies similar to the present investigation have already been conducted, especially in human tumors, and it must be said that they are less than novel.

Comments on the Quality of English Language

English used in this manuscript is acceptable.

Author Response

Thank you for your insightful comments regarding our paper. Below, we have addressed your points in more detail:

  • Detection of mitotic figures in specific areas: Our models were only trained on patches that contained mitoses marked by pathologists. However, we acknowledge that the performance could vary across different sections of tissue. For further discussion we propose expanding our dataset to include a broader range of areas, including those not initially marked by pathologists as well as annotations labelling areas containing both tumour and non-tumour tissue, to evaluate and enhance our model's generalisability.
    • Added to the discussion: “It must be further denoted that the methodology is applied to only patches containing mitosis that were annotated by the collaborating pathologists. Therefore, we propose expanding our dataset to include a broader range of sections, including those not initially marked by pathologists, to evaluate and enhance our model's generalisability. The data should include labels for areas containing tumour and non-tumour tissue to fully consider the overall impact of this mitosis detection method.”

  • Clinical application of the F1-Score: The F1-score achieved in our study demonstrates a promising level of accuracy in the canine domain, but we agree that it is crucial to at least discuss its clinical relevance. To address this, in the discussion, we propose further work where pathologists could use our model as an assistive tool and determine the practical effectiveness and reliability in a veterinary clinical setting.
    • Added to the discussion: “While our F1-score demonstrates competitive performance for detecting mitosis in the canine domain, the clinical relevance and applicability of this metric should be taken into account. Future work should focus on employing this method as a supportive tool, assessing its practical effectiveness and reliability in a veterinary clinical setting.”

  • Applying this method to other tumour types: Our study focuses on mitosis in cPWT, however, we could potentially adapt this work to other cSTS subtypes and other tumour types. Future research could involve directly applying our trained model or retraining/ fine-tuning our models with alternative datasets, to evaluate the adaptability and effectiveness of our approach in broader oncological contexts.

  • Added to the discussion: “Our focus for this study is on cPWT, however, we could potentially adapt this method to other cSTS subtypes as well as to other tumour types. An additional study might explore the application of cPWT-trained models to different cSTS subtypes to assess if comparable outcomes are achieved. Nevertheless, given that tumour types from various domains exhibit unique challenges due to their specific histological characteristics, it may be necessary to train or fine-tune models using tumour specific datasets to evaluate the efficacy of this approach.”

Reviewer 3 Report

Comments and Suggestions for Authors

In this article authors attempted to automate mitosis detection in canine perivascular wall tumors by keeping pathologists in the loop and adjusting relative f1-score. I agree with authors, mitosis count is subjected to inter and intra predictor variability. Automation of mitosis detection is of utmost importance as it’s a laborious work. Authors used 2 step annotation process. In step 1 authors used 2 blinded individual veterinary pathologists mitosis count and using this authors pre trained Faster R-CNN model. In the next step authors adjusted Faster R-CNN by updating dataset reviewed by pathologists for false positives with outcome from pretrained dataset. Authors achieved F-score of 0.75 which authors propose as competitive and state of art in canine mitosis domain. With initial dataset sensitivity was good but precision was not great, meaning higher number of false positives being detected. With updated dataset sensitivity was compromised slightly but precision has increased quite significantly bringing average F1-score to 0.75. Overall, this manuscript is well written. Although there are many mitosis detection tools available, canine perivascular wall tumors specific mitosis detection this is first. In my opinion manuscript can be improved if Table A6, A9 and A10 in moved to results section.

Finally, Table A10 legend abruptly ended at “Both the”, please rectify this typing error.

Author Response

Thank you for your time to read the manuscript and your comments. We have taken into account your suggestions and have moved tables tables A6 and A9 to the results section for better readability. However, we believe that Table A10 is better placed in the Appendix as it discusses a methodology and specific results that do not align directly with the main content of this paper. Therefore, for clarity and to focus on our main method, these results from Table A10 (now presented as Table A8) are better placed in the Appendix.

Round 2

Reviewer 2 Report

Comments and Suggestions for Authors

It is my opinion that additional analyses should have been added to this study. Still, I am disappointed that the authors only added a note to the Discussion as "to be done in the future."